# Contrastive Mutual Information Maximization for Binary Neural Networks

## Abstract

Neural network binarization accelerates deep models by quantizing their weights and activations into 1-bit. However, there is still a huge performance gap between Binary Neural Networks (BNNs) and their full-precision counterparts. As the quantization error caused by weights binarization has been reduced in earlier works, the activations binarization becomes the major obstacle for further improvement of the accuracy. In spite of studies about the full-precision networks highlighting the distributions of activations, few works study the distribution of the binary activations in BNNs. In this paper, we introduce mutual information as the metric to measure the information shared by the binary and the latent full-precision activations. Then we maximize the mutual information by establishing a contrastive learning framework while training BNNs. Specifically, the representation ability of the BNNs is greatly strengthened via pulling the positive pairs with binary and full-precision activations from the same input samples, as well as pushing negative pairs from different samples (the number of negative pairs can be exponentially large). This benefits the downstream tasks, not only classification but also segmentation and depth estimation, *etc*. The experimental results show that our method can be implemented as a pile-up module on existing state-of-the-art binarization methods and can remarkably improve the performance over them on CIFAR-10/100 and ImageNet, in addition to the good generalization ability on NYUD-v2.

## 1 Introduction

Although deep neural networks (DNNs) [1] have achieved remarkable success in various computer vision tasks such as image classification [2] and semantic image segmentation [3], their over-parametrization problem makes them computationally expensive and storage excessive. To advance the development of deep learning towards resource-constrained edge devices, researchers proposed several neural network compression paradigms, such as network pruning [4, 5], knowledge distillation [6] and network quantization [7, 8]. Among the network quantization methods, the network binarization method [7] stands out for quantizing weights and activations (*i.e.* intermediate feature maps) to $\pm 1$, compressing the full-precision counterpart $32\times$, and replacing time-consuming inner-product in full-precision networks with efficient xnor-bitcount operation in the BNNs.

However, severe accuracy drops always exist between full-precision models and their binary counterparts. To tackle this problem, previous works mainly focus on reducing the quantization error induced by weights binarization [9, 10], and elaborately approximating binarization function to relieve the gradient mismatch in the backward propagation [11, 8]. Indeed, they achieve the state-of-the-art performance. Yet with those two paradigms developing, narrowing down the quantization error and enhancing the gradient transmission reach their bottlenecks [12, 13], since the 1W32A (only quantizing the weights into 1-bit, remaining the activations 32-bit) models are capable of performing as well as the full-precision models, implying that the activations binarization becomes the main issue for further performance improvement.

To address the accuracy degradation caused by the activations binarization, a few studies propose to regulate the distributions of the binary activations, *e.g.* researchers in [14] design a distribution loss to explicitly regularize the activation flow; researchers in [13] propose to shift the thresholds of binary activation functions to make the distribution of binary activation unbalanced. They heuristically design low-level patterns to analyze the distributions of binary activations such as minimum of the

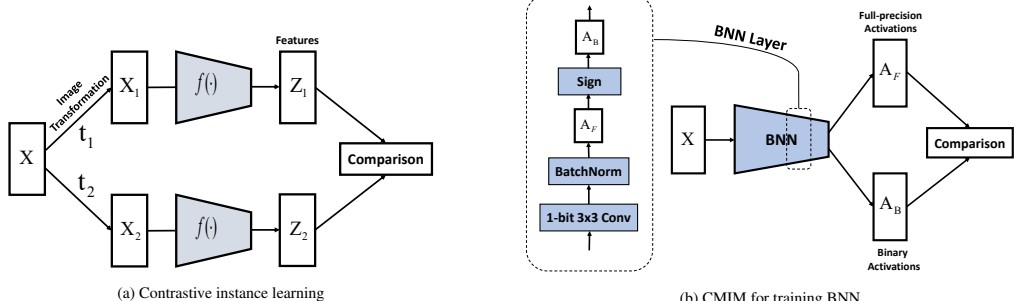

Figure 1: **(a)**: In contrastive instance learning, the features from different transformations of the same input image are compared to each other. **(b)**: However BNN can yield the binary activations $A_B$ and full-precision activations $A_F$ (*i.e.* two transformations of an image both from the same BNN) in the same forward pass, thus the BNN can act as two image transformations in the literature of contrastive learning.

activations and the balanced property of distributions. Nevertheless, they neglect the high-level indicators of the distribution and the unique characteristics of BNN, where the binary activations and latent full-precision activations exist in the same forward pass. Thus, we argue that the high-level properties of distributions, such as correlations and dependencies between binary and full-precision activations should be captured and utilized.

In this work, we explore introducing mutual information for BNNs, in which the mutual information acts as a fundamental quantity to measure the information amount shared by the binary and latent real-valued activations in BNNs. In contrast to the aforementioned works focusing on learning the distribution of binary activations, mutual information naturally captures non-linear statistical dependencies between variables, and thus can be used as a measure of true dependence [15]. Based on this metric, we propose a novel method, termed as **C**ontrastive **M**utual **I**nformation **M**aximization for Binary Neural Networks (**CMIM**-BNN). Specifically, we design a highly effective optimization strategy using contrastive estimation for the mutual information maximization. As illustrated in Figure 1, we replace the data transformation module in contrastive learning with the exclusive structure in BNNs, where full-precision and binary activations are in the same forward pass. In this way, contrastive learning contributes to inter-class decorrelation of binary activations, and avoids collapse solutions. In other words, our method is built upon a contrastive learning framework to learn representative binary activations, in which we pull the binary activation closer to the full-precision activation and push the binary activation further away from other binary activations in the contrastive space. Moreover, by utilizing an additional MLP module to extract representations of activations, our method can explicitly capture higher-order dependencies in the contrastive space. To the best of our knowledge, it is the first work aiming at maximizing the mutual information of the activations in BNNs within a contrastive learning framework.

Overall, the contributions of this paper are three-fold:

- Considering the distributions of activations, we propose a novel CMIM framework to optimize BNNs, by maximizing the mutual information between the binary activation and its latent real-valued counterpart;
- We develop an effective contrastive learning strategy to achieve the goal of mutual information maximization for BNNs, and benefited from it, the representation ability of BNNs is clearly strengthened for not only the classification task but also downstream CV tasks;
- Experimental results show that our method can significantly improve the existing SOTA methods over the classification task on CIFAR-10/100 and ImageNet, *e.g.* 6.4% on CIFAR-100 and 3.0% on ImageNet. Besides, we also demonstrate good generalization ability of the proposed CMIM on other challenging CV tasks such as depth estimation and semantic segmentation.

## 2 MUTUAL INFORMATION MAXIMIZATION FOR TRAINING BNNS

### 2.1 PRELIMINARIES

We first define a Multi-Layer Perceptron (MLP) with $K$ layers. For simplification of derivation, we discard the bias term of the network. Then the MLP $f(\mathbf{x})$ can be denoted as:

$$f(\mathbf{W}^1, \cdots, \mathbf{W}^K; \mathbf{x}) = (\mathbf{W}^K \cdot \sigma \cdot \mathbf{W}^{K-1} \cdot \cdots \cdot \sigma \cdot \mathbf{W}^1)(\mathbf{x}), \tag{1}$$

where $\mathbf{x}$ is the input sample and $\mathbf{W}^k : \mathbb{R}^{d_{k-1}} \longmapsto \mathbb{R}^{d_k} (k = 1, ..., K)$ stands for the weight matrix connecting the $(k-1)$-th and the $k$-th layer, with $d_{k-1}$ and $d_k$ representing the sizes of the input and output of the $k$-th network layer, respectively. The $\sigma(\cdot)$ function performs element-wise activation for the feature maps. Notably, for a convolution layer with the input map of $m$ channels and the output map of $n$ channels, and the size of the kernel $w \times h$, it results in $m \times n \times w \times h$ parameters. We can re-arrange the parameters to a weight matrix of size $n \times (m \times h \times w)$, such that this convolution layer can also operate in the same way as the other fully-connected layers. Hence, it is sufficient to consider networks with the fully-connected layers.

Based on those predefined notions, the sectional MLP $f^k(\mathbf{x})$ with the front $k$ layers of the $f(\mathbf{x})$ can be represented as:

$$f^k(\mathbf{W}^1, \cdots, \mathbf{W}^k; \mathbf{x}) = (\mathbf{W}^k \cdot \sigma \cdots \sigma \cdot \mathbf{W}^1)(\mathbf{x}). \tag{2}$$

And the MLP $f$ can be seen as a special case in the function sequence $\{f^k\}(k \in \{1, \cdots, K\})$, *i.e.* $f = f^K$, when $k = K$.

**Binary Neural Networks.** Here, we revisit the general binarization method in [16, 7], which maintains full-precision latent variables $\mathbf{W}_F$ for gradient updates, and the $k$-th weight matrix $\mathbf{W}_F^k$ is binarized into $\pm 1$, obataining the binary weight matrix $\mathbf{W}_B^k$ by a binarize function (normally $sgn(\cdot)$), *i.e.* $\mathbf{W}_B^k = sgn(\mathbf{W}_F^k)$. Then the intermediate activation map (full-precision) of the $k$-th layer is produced by $\mathbf{A}_F^k = \mathbf{W}_B^k \mathbf{A}_B^{k-1}$, then the same quantization method is used to binarize the full-precision activation map as $\mathbf{A}_B^k = sgn(\mathbf{A}_F^k)$, and a whole forward pass of binarization is performed by iterating this process for $L$ times.

**Mutual Information.** For two discrete variables $\mathbf{X}$ and $\mathbf{Y}$, their mutual information can be defined as [17]:

$$I(\mathbf{X}, \mathbf{Y}) = \sum_{x,y} P_{\mathbf{XY}}(x, y) \log \frac{P_{\mathbf{XY}}(x, y)}{P_{\mathbf{X}}(x) P_{\mathbf{Y}}(y)}, \tag{3}$$

where $P_{\mathbf{XY}}(x, y)$ is the joint distribution, $P_{\mathbf{X}}(x) = \sum_y P_{\mathbf{XY}}(x, y)$ and $P_{\mathbf{Y}}(y) = \sum_x P_{\mathbf{XY}}(x, y)$ are the marginals of $\mathbf{X}$ and $\mathbf{Y}$, respectively.

Mutual information quantifies the amount of information obtained about one random variable by observing the other random variable. It is a dimensionless quantity with (generally) units of bits, and can be thought as the reduction in uncertainty about one random variable given knowledge of another. High mutual information indicates a large reduction in uncertainty; low mutual information indicates a small reduction; and zero mutual information between two random variables means the variables are independent. In the content of binarization, considering the binary and full-precision activations as random variables, we would like them share as much information as possible, since the binary activations are proceeded from their corresponding full-precision activations. Theoretically, the mutual information between those two variables should be maximized.

## 2.2 CONTRASTIVE MUTUAL INFORMATION MAXIMIZATION

In the following section, we formalize the idea of constructing a Noise-Contrastive Estimation (NCE) loss to maximize the mutual information between the binary and the full-precision activations. Particularly, we derive a novel CMIM loss for training BNNs, where NCE is introduced to avoid the direct mutual information computation by estimating it with its lower bound in Eq. 7.

For binary network $f_B$ and its latent full-precision counterpart $f_F$ in the same training iteration, the series of their activations $\{\mathbf{a}_B^k\}$ and $\{\mathbf{a}_F^k\}(k \in \{1, \cdots, K\})$, where $\mathbf{A}_B^k = (\mathbf{a}_B^{k,1}, \cdots, \mathbf{a}_B^{k,N})$ and $\mathbf{A}_F^k = (\mathbf{a}_F^{k,1}, \cdots, \mathbf{a}_F^{k,N})$ can be considered as a series of variables. The corresponding variables $(\mathbf{a}_B^k, \mathbf{a}_F^k)$ should share more information, *i.e.* the mutual information of the same layer's output activations $I(\mathbf{a}_B^k, \mathbf{a}_F^k)(k \in \{1, \cdots, K\})$ should be maximized to enforce them mutually dependent.

To this end, we introduce the contrastive learning framework into our targeted binarization task. The basic idea of contrastive learning is to compare different views of the data (usually under different data augmentations) to calculate similarity scores [18, 19, 20, 21, 22]. This framework is suitable for our case, since the binary and full-precision activations can be seen as two different views.

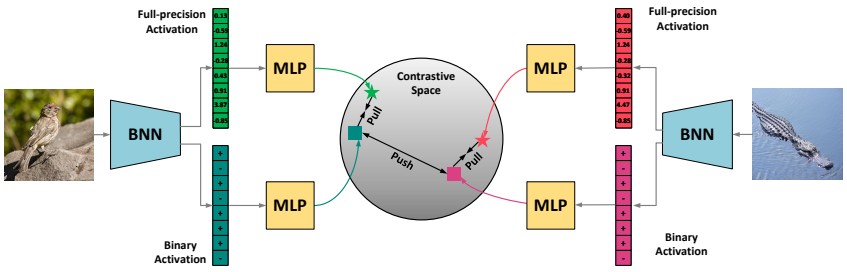

Figure 2: Feeding two images into a BNN, and obtaining the three pairs of binary and full-precision activations. Our goal is to embed the activations into a contrastive space, then learn from the pair correlation with the contrastive learning task in Eq. 9.

For a training batch with $N$ samples, the samples can be denoted as: $\{\mathbf{x}_i\}(i \in \{1, \cdots, N\})$. We feed a batch of samples to the BNN and obtain $KN^2$ pairs of activations $(\mathbf{a}_B^{k,i}, \mathbf{a}_F^{k,j})$, which augments the data for the auxiliary task. We define a pair containing two activations from the same sample as positive pair, *i.e.* if $i = j$, $(\mathbf{a}_B^{k,i}, \mathbf{a}_F^{k,j})_+$ and *vice versa*. With the Bayes' theorem, the posterior probability of two activations from the positive pair can be formalized as:

$$P(i = j \mid \mathbf{a}_B^{k,i}, \mathbf{a}_F^{k,j}) = \frac{P(\mathbf{a}_B^{k,i}, \mathbf{a}_F^{k,j} \mid i = j)\frac{1}{N}}{P(\mathbf{a}_B^{k,i}, \mathbf{a}_F^{k,j} \mid i = j)\frac{1}{N} + P(\mathbf{a}_B^{k,i}, \mathbf{a}_F^{k,j} \mid i \neq j)\frac{N-1}{N}}. \tag{4}$$

And the probability of activations from negative pair is: $P(i \neq j \mid \mathbf{a}_B^{k,i}, \mathbf{a}_F^{k,j}) = 1 - P(i = j \mid \mathbf{a}_B^{k,i}, \mathbf{a}_F^{k,j})$. To simplify the NCE derivative, several works [23, 24, 25] build assumption about the dependence of the variables, we also use this assumption that the activations from positive pairs are dependent and the ones from negative pairs are independent, *i.e.* $P(\mathbf{a}_B^{k,i}, \mathbf{a}_F^{k,j} \mid i = j) = P(\mathbf{a}_B^{k,i}, \mathbf{a}_F^{k,j})$ and $P(\mathbf{a}_B^{k,i}, \mathbf{a}_F^{k,j} \mid i \neq j) = P(\mathbf{a}_B^{k,i})P(\mathbf{a}_F^{k,j})$. Hence, the above equation can be simplified as:

$$P(i = j \mid \mathbf{a}_B^{k,i}, \mathbf{a}_F^{k,j}) = \frac{P(\mathbf{a}_B^{k,i}, \mathbf{a}_F^{k,j})}{P(\mathbf{a}_B^{k,i}, \mathbf{a}_F^{k,j}) + P(\mathbf{a}_B^{k,i})P(\mathbf{a}_F^{k,j})(N-1)}. \tag{5}$$

Performing logarithm to Eq. 5 and arranging the terms, we can achieve

$$\log P(i = j \mid \mathbf{a}_B^{k,i}, \mathbf{a}_F^{k,j}) = -\log\left[1 + (N-1)\frac{P(\mathbf{a}_B^{k,i})P(\mathbf{a}_F^{k,j})}{P(\mathbf{a}_B^{k,i}, \mathbf{a}_F^{k,j})}\right] \leq \log\frac{P(\mathbf{a}_B^{k,i}, \mathbf{a}_F^{k,j})}{P(\mathbf{a}_B^{k,i})P(\mathbf{a}_F^{k,j})} - \log(N-1). \tag{6}$$

Taking expectation on both sides with respect to $P(\mathbf{a}_B^{k,i}, \mathbf{a}_F^{k,j})$, and combining the definition of mutual information in Eq. 3, we can derive the form of mutual information as:

$$\overbrace{I(\mathbf{a}_B^k, \mathbf{a}_F^k)}^{\text{targeted mutual information}} = \sum_i \sum_j P(\mathbf{a}_B^{k,i}, \mathbf{a}_F^{k,j}) \log \frac{P(\mathbf{a}_B^{k,i}, \mathbf{a}_F^{k,j})}{P(\mathbf{a}_B^{k,i})P(\mathbf{a}_F^{k,j})}$$

$$\geq \sum_i \sum_j P(\mathbf{a}_B^{k,i}, \mathbf{a}_F^{k,j} \mid i = j)\left[\log P(i = j \mid \mathbf{a}_B^{k,i}, \mathbf{a}_F^{k,j}) + \log(N-1)\right] \tag{7}$$

$$= \overbrace{\mathbb{E}_{P(\mathbf{a}_B^{k,i}, \mathbf{a}_F^{k,j} \mid i=j)}\left[\log P(i = j \mid \mathbf{a}_B^{k,i}, \mathbf{a}_F^{k,j})\right]}^{\text{optimized lower bound}} + \log(N-1),$$

where $I(\mathbf{a}_B^k, \mathbf{a}_F^k)$ is the mutual information between the binary and full-precision distributions, our targeted object. Instead of directly maximizing the mutual information, we choose to maximize its lower bound in the Eq. 7. However, the distribution $P(i = j \mid \mathbf{a}_B^{k,i}, \mathbf{a}_F^{k,j})$ is hard to estimate. We take advantage of the idea of contrastive learning, and introduce a critic function $h$ to approximate the targeted distribution [18, 19, 20]. In practice, we use the following:

$$h(\mathbf{a}_B^{k,i}, \mathbf{a}_F^{k,j}) = \frac{\exp(\tau(\mathbf{a}_B^{k,i})^\top \mathbf{a}_F^{k,j})}{\sum_j \exp(\tau(\mathbf{a}_B^{k,i})^\top \mathbf{a}_F^{k,j})} \tag{8}$$

---

**Algorithm 1** Forward and Backward Propagation of CMIM

**Require:** A minibatch of data samples $(\mathbf{X}, \mathbf{Y})$, current binary weight $\mathbf{W}_B^k$, latent full-precision weights $\mathbf{W}_F^k$, and learning rate $\eta$.

**Ensure:** Update weights $\mathbf{W}_F^{k}{}'$.

1: **Forward Propagation**:
2: **for** $k = 1$ to $K - 1$ **do**
3:    Binarize latent weights:  $\mathbf{W}_B^k \leftarrow$ sgn$(\mathbf{W}_F^k)$;
4:    Perform binary operation with the activations of next layer: $\mathbf{A}_F^k \leftarrow$ XnorDotProduct$(\mathbf{W}_B^k, \mathbf{A}_B^{k-1})$;
5:    Perform Batch Normalization: $\mathbf{A}_F^k \leftarrow$ BatchNorm$(\mathbf{A}_F^k)$;
6:    Binarize full-precision activations and obtain binary activations : $\mathbf{A}_B^k \leftarrow$ sgn$(\mathbf{A}_F^k)$;
7: **end for**
8: For $k = 1, \cdots, K$, pair $\left\{\mathbf{a}_B^{k,i}\right\}$ and $\left\{\mathbf{a}_B^{k,j}\right\}$ as negative and positive pairs, then use Eq. 9 layer by layer to compute the NCE loss $\mathcal{L}_{NCE}^k$ between $\mathbf{A}_B^k$ and $\mathbf{A}_F^k$ for contrastive learning;
9: Combine a series of NCE loss $\left\{\mathcal{L}_{NCE}^k\right\}$ with the classification loss $\mathcal{L}$ into the CMIM loss $\mathcal{L}_{CMIM}$, with Eq. 11;
10: **Backward Propagation**: compute the gradient of the overall loss function, *i.e.* $\frac{\partial \mathcal{L}}{\partial \mathbf{W_B}}$, using the straight through estimator to tackle the sign function;
11: **Parameter Update**: update the full-precision weights: $\mathbf{W}_F^{i}{}' \leftarrow \mathbf{W}_F^k - \eta \frac{\partial \mathcal{L}}{\partial \mathbf{W}_B^k}$.

---

where $\tau$ is a temperature parameter that controls the concentration level of the distribution [6]. $\tau$ is important for supervised feature learning, and also necessary for tuning the concentration of $\mathbf{a}_B^{k,i}$ and $\mathbf{a}_F^{k,j}$ on our contrastive space.

**Loss Function.** We define the contrastive loss function $\mathcal{L}_{NCE}^k$ between the $k$-th layer's activations $\mathbf{A}_B^k$ and $\mathbf{A}_F^k$ as:

$$\mathcal{L}_{NCE}^k = \mathbb{E}_{P(\mathbf{a}_B^{k,i}, \mathbf{a}_F^{k,j}|i=j)}\left[\log h(\mathbf{a}_B^{k,i}, \mathbf{a}_F^{k,j})\right] + N\mathbb{E}_{P(\mathbf{a}_B^{k,i}, \mathbf{a}_F^{k,j}|i\neq j)}\left[\log(1 - h(\mathbf{a}_B^{k,i}, \mathbf{a}_F^{k,j}))\right]. \quad (9)$$

We would like to comment on the above loss function from the perspective of contrastive learning. The first term of positive pairs is optimized for capturing more intra-class correlations and the second term of negative pairs is for inter-class decorrelation. Because the pair construction is instance-wise, the number of negative samples theoretically can be the size of the entire training set, *e.g.* 1.2 million for ImageNet. With those additional hand-craft designed contrastive pairs for the proxy optimization problem in Eq. 9, the representation capacity of BNNs can be further improved, as many contrastive learning methods demonstrated [22, 18, 19, 20].

Moreover, the optimal $\hat{h} = \arg\max_h \mathcal{L}_{NCE}^k$ can approximate the targeted distribution, *i.e.*

$$\hat{h}(\mathbf{a}_B^{k,i}, \mathbf{a}_F^{k,j}) = P(i = j \mid \mathbf{a}_B^{k,i}, \mathbf{a}_F^{k,j}), \quad (10)$$

where the detailed proof is shown in the supplementary material. Thus, with Eq. 7-10 we can derive that minimizing the NCE loss $\mathcal{L}_{NCE}^k$ is equivalent to maximizing the targeted mutual information between the binary and full-precision activations, $I(\mathbf{a}_B^k, \mathbf{a}_F^k)$.

Combining the series of NCE loss from different layers $\left\{\mathcal{L}_{NCE}^k\right\}, (k = 1, \cdots, K)$, the overall loss $\mathcal{L}$ can be defined as:

$$\mathcal{L} = \lambda \sum_{k=1}^K \frac{\mathcal{L}_{NCE}^k}{\beta^{K-1-k}} + \mathcal{L}_{cls}, \quad (11)$$

where $\mathcal{L}_{cls}$ is the classification loss respect to the ground truth, $\lambda$ is used to control the degree of NCE loss, $\beta$ is a coefficient greater than 1, and we denote the CMIM loss as $\mathcal{L}_{CMIM} = \sum_{k=1}^K \frac{\mathcal{L}_{NCE}^k}{\beta^{K-1-k}}$.

Hence, the $\beta^{K-1-k}$ decreases with $k$ increasing and consequently the $\frac{\mathcal{L}_{NCE}^k}{\beta^{K-1-k}}$ increases. In this way, the activations of latter layer can be substantially retained, which leads to better performance in practice. The complete training process of CMIM is presented in Algorithm 1.

**Discussion on the CMIM Loss.** Besides the theoretical formulation from the perspective of mutual information maximization, we also provide an intuitive explanation about CMIM. As illustrated in Figure 2, we strengthen the representation ability of binary activations (Figure 3) via designing a proxy task with the contrastive learning framework. By embedding the activations to the contrastive

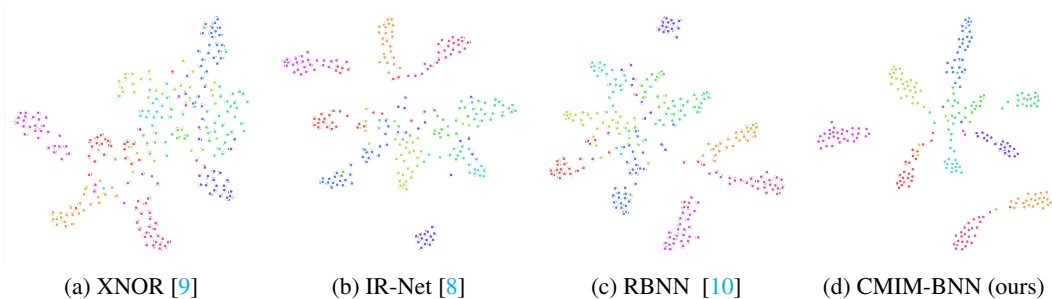

| (a) XNOR [9] | (b) IR-Net [8] | (c) RBNN [10] | (d) CMIM-BNN (ours) |

Figure 3: t-SNE [26] visualization of the activations representing for random 10 classes in CIFAR-100. Every color represents a different class. We can clearly witness the improvement of our method for learning better binary representations.

space and pull-and-push the paired embeddings, the BNNs can learn better representations from this difficult yet effective auxiliary contrastive learning task. Note that even though we only pick up two images to formulate Figure 2, the actual number of negative samples can be huge in practice (*e.g.* 16,384 for training ResNet-18 on ImageNet), benefit from the MemoryBank [24] technique.

With this property, we speculate that the contrastive pairing works as the data augmentation, which contributes to our method. This additional pairing provides more information for training the BNNs, thus our CMIM loss can be treated as an overfitting-mitigated module. We also conduct experiments in the Section 3.2 and 3.3 to validate our speculation.

**Difference with other contrastive learning methods.** The key idea of contrastive learning is to pull representations close in positive pairs and push representations apart in negative pairs in a contrastive space. Several self-supervised learning methods are rooted in well-established idea of the mutual information maximization, such as Deep InfoMax [19], Contrastive Predictive Coding [18], MemoryBank [24], Augmented Multiscale DIM [20], MoCo [21] and SimSaim [22]. These are based on NCE [23] and InfoNCE [19] which can be seen as a lower bound on mutual information [27].

The formulation of our CMIM-BNN is similar to the classic contrastive learning methods, where we all are inspired by NCE. However, our approach has several differences from those methods. Firstly, the training process of BNNs is different from regular network training, where binary and latent full-precision activations exist in the same forward pass. We seamlessly integrate this mixed-activation property with NCE, and thus the targeted lower bound formulated for optimization is different. Secondly, the binary and full-precision weights are both optimized by the NCE loss (*i.e.* the two view augmentation networks are optimized simultaneously in contrastive learning), yet most aforementioned contrastive learning methods optimize their view augmentation networks separately.

## 3 EXPERIMENTS

In this section, we first conduct experiments to compare with existing state-of-the-art methods in image classification. Following popular settings in most studies, we use CIFAR-10/100 [28] and ImageNet ILSVRC-2012 [29] to validate the effectiveness of our proposed binarization method. Besides comparing our method with the SOTA methods, we design experiments in semantic segmentation and depth estimation tasks on the NYUD-v2 [30] dataset to testify the generalization ability of our method. Meanwhile, we design a series of ablation studies to verify the effectiveness of our proposed technique, and we empirically explain the efficacy of CMIM from the perspective of mitigating overfitting. All experiments are implemented using PyTorch [31] with one NVIDIA RTX 6000 GPU while training on CIFAR-10/100 and NYUD-v2, and two GPUs on ImageNet.

**Experimental Setup.** On CIFAR-10/100, the BNNs are trained by CMIM for 400 epochs with batch size of 256, initial learning rate of 0.1 and cosine learning rate scheduler. We adopt SGD optimizer with momentum of 0.9 and weight decay of 1e-4. On ImageNet, binary models are trained for 100 epochs with batch size of 256. SGD optimizer is applied with momentum of 0.9, weight decay of 1e-4, initial learning rate of 0.1 with cosine learning rate scheduler.

Table 1: Top-1 accuracy (%) on CIFAR-10 (C-10) and CIFAR-100 (C-100) test set. The higher the better. W/A denotes the bit number of weights/activations.

| Topology | Method | Bit-width (W/A) | Acc.(%) (C-10) | Acc.(%) (C-100) |
|---|---|---|---|---|
| ResNet -20 | Full-precision | 32/32 | 92.1 | 70.7 |
| | DoReFa [32] | 1/1 | 79.3 | - |
| | QSQ [33] | 1/1 | 84.1 | - |
| | SLB [34] | 1/1 | 85.5 | - |
| | LNS [35] | 1/1 | 85.8 | - |
| | IR-Net [8] | 1/1 | 86.5 | 65.6 |
| | RBNN [10] | 1/1 | 87.0 | 66.0 |
| | IR-Net + CMIM | 1/1 | **87.3** | **68.1** |
| | RBNN + CMIM | 1/1 | **87.6** | **68.2** |
| ResNet -18 | Full-precision | 32/32 | 93.0 | 72.5 |
| | RAD [14] | 1/1 | 90.5 | - |
| | Proxy-BNN [36] | 1/1 | 91.8 | 67.2 |
| | IR-Net [8] | 1/1 | 91.6 | 64.5 |
| | RBNN [10] | 1/1 | 92.2 | 65.3 |
| | IR-Net + CMIM | 1/1 | **92.2** | **71.2** |
| | RBNN + CMIM | 1/1 | **92.8** | **71.4** |
| VGG -small | Full-precision | 32/32 | 94.1 | 73.0 |
| | XNOR [9] | 1/1 | 90.5 | - |
| | DoReFa [32] | 1/1 | 90.2 | - |
| | RAD [14] | 1/1 | 90.5 | - |
| | QSQ [33] | 1/1 | 90.0 | - |
| | SLB [34] | 1/1 | 92.0 | - |
| | Proxy-BNN [36] | 1/1 | 91.8 | 67.2 |
| | IR-Net [8] | 1/1 | 90.4 | 67.0 |
| | RBNN [10] | 1/1 | 91.3 | 67.4 |
| | IR-Net + CMIM | 1/1 | **92.0** | **70.0** |
| | RBNN + CMIM | 1/1 | **92.2** | **71.0** |

Table 2: Top-1 and Top-5 accuracy on ImageNet. † represents the architecture which varies from the standard ResNet architecture but in the same FLOPs level.

| Topology | Method | BW (W/A) | Top-1 (%) | Top-5 (%) |
|---|---|---|---|---|
| ResNet-18 | Full-precision | 32/32 | 69.6 | 89.2 |
| | ABC-Net [37] | 1/1 | 42.7 | 67.6 |
| | XNOR-Net [9] | 1/1 | 51.2 | 73.2 |
| | BNN+ [7] | 1/1 | 53.0 | 72.6 |
| | DoReFa [32] | 1/2 | 53.4 | - |
| | XNOR++ [38] | 1/1 | 57.1 | 79.9 |
| | BiReal [11] | 1/1 | 56.4 | 79.5 |
| | IR-Net [8] | 1/1 | 58.1 | 80.0 |
| | RBNN [10] | 1/1 | 59.9 | 81.0 |
| | BiReal + CMIM | 1/1 | **60.1** | **81.3** |
| | IR-Net + CMIM | 1/1 | **61.2** | **83.0** |
| | RBNN + CMIM | 1/1 | **62.5** | **84.2** |
| | BATS [39]† | 1/1 | 60.4 | 83.0 |
| | BATS + CMIM† | 1/1 | **63.0** | **85.1** |
| | ReActNet [40]† | 1/1 | 69.4 | 85.5 |
| | ReActNet + CMIM† | 1/1 | **71.0** | **86.3** |
| ResNet-34 | Full-precision | 32/32 | 73.3 | 91.3 |
| | ABC-Net [37] | 1/1 | 52.4 | 76.5 |
| | XNOR-Net [9] | 1/1 | 53.1 | 76.2 |
| | BiReal [11] | 1/1 | 62.2 | 83.9 |
| | XNOR++ [38] | 1/1 | 57.1 | 79.9 |
| | IR-Net [8] | 1/1 | 62.9 | 84.1 |
| | LNS [35] | 1/1 | 59.4 | 81.7 |
| | RBNN [10] | 1/1 | 63.1 | 84.4 |
| | IR-Net + CMIM | 1/1 | **64.9** | **85.8** |
| | RBNN + CMIM | 1/1 | **65.0** | **85.7** |

## 3.1 EXPERIMENTAL RESULTS

**CIFAR-10/100** are widely-used image classification datasets, where each consists of 50K training images and 10K testing images of size 32×32 divided into 10/100 classes. 10K training images are randomly sampled for cross-validation and the rest images are utilized for training. Data augmentation strategy includes random crop and random flipping as in [41] during training.

For ResNet-20, we compare with DoReFa [32], QSQ [33], SLB [34], LNS [35], IR-Net [8] and RBNN [10]. For ResNet-18, RAD [14], Proxy-BNN [36], IR-Net and RBNN are chosen to be the benchmarks. For VGG-small, our method is compared with IR-Net and RBNN, *etc*.

As presented in Table 1, CMIM constantly outperforms other SOTA methods. On CIFAR-100, our method achieves 2.5%, 6.1% and 4.0% performance improvement with ResNet-20, ResNet-18 and VGG-small architectures, respectively. To show the pile-up property , we add CMIM on different baseline methods, and we can obviously observe the accuracy gain with CMIM.

**ImageNet** is a dataset with 1.2 million training images and 50k validation images equally divided into 1K classes. ImageNet has greater diversity, and its image size is 469×387 (average). We report the single-crop evaluation result using 224×224 center crop from images.

For ResNet-18, we compare our method with XNOR-Net [9], ABC-Net [37], DoReFa [32], BiReal [11], XNOR++ [38], IR-Net [8], RBNN [10]. For ResNet-34, we compare our method with BiReal, IR-Net and RBNN, *etc.*. All experimental results are either taken from their published papers or reproduced ourselves using their code. As demonstrated in Table 2, our proposed method exceeds all the methods in both top-1 and top-5 accuracy. Particularly, CMIM achieves around 2.5% Top-1 accuracy gain with ResNet-18 architecture, as well as 1.9% Top-1 accuracy improvement with ResNet-34 architecture, compared with the state-of-the-art method RBNN.

Apart from the aforementioned methods binarizing networks with the ResNet architecture, we also conduct experiments to compare with methods designing BNNs with variant architectures. For example, BATS [39] utilizes neural architecture search [42] to automatically design a BNN architecture and ReActNet [40] designs BNNs with mobile-net [43] architectures. With adding CMIM module on those architectures, we observe that BNNs trained with CMIM have noticeable performance gain, which further consolidates the effectiveness of our method.

## 3.2 NUMBER OF NEGATIVE SAMPLES IN CMIM

The number of negative samples n_nce is an important hyper-parameter in our method, which ensures the estimation accuracy level of the optimized distribution in Eq. 7. We perform experiments with

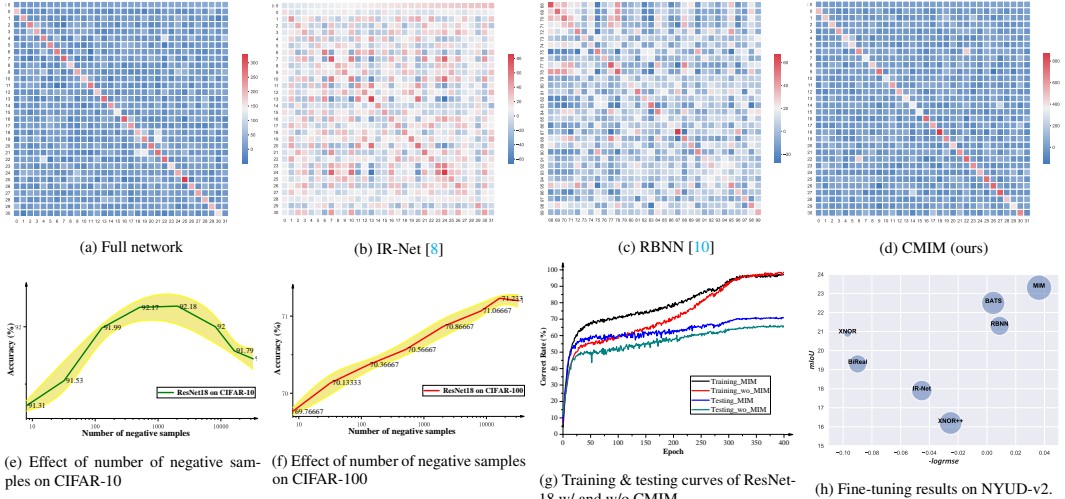

(a) Full network      (b) IR-Net [8]      (c) RBNN [10]      (d) CMIM (ours)

(e) Effect of number of negative samples on CIFAR-10

(f) Effect of number of negative samples on CIFAR-100

(g) Training & testing curves of ResNet-18 w/ and w/o CMIM

(h) Fine-tuning results on NYUD-v2.

Figure 4: In-depth analysis on different aspects of the proposed approach including a comparison of learned correlation maps from different methods (a-d), the effect of number of negative samples in contrastive mutual information maximization (c-f), training and testing curves (g), and a comparison of fine-tuning results (h).

ResNet18 on CIFAR-100 for parameter analysis of n_nce, with range from $2^0$ to $2^{15}$. As the results in Figure 4f presented, the accuracy arises with increasing n_nce, which also validates our speculation in the Section 2.2 that the contrastive pairing module, serving as a data augmentation module in training, contributes to the performance improvement of CMIM.

### 3.3 MITIGATE OVERFITTING

A good training objective should be reflected in consistent improvement in the testing performance. We investigate the relation between the training and the testing accuracy across iterations. Figure 4g shows that (1) the binary ResNet-18 can reach 100% on training set of CIFAR-100, which means its representative ability is enough for this dataset; (2) on the final stage, the testing performance of the BNN trained with CMIM loss is mush better, while the training performance is relatively lower, which is a clear sign of mitigating overfitting. In addition, as the results shown in the Table 3, we can observe the phenomenon that the accuracy gain on CIFAR-100 is more noticeable than the gain on ImageNet. This phenomenon can also be explained from the perspective of mitigating overfitting. Since the contrastive pairing (data augmentation for the proxy contrastive learning task) plays a significant role in improving the performance of BNNs, and the data for training is sufficient on ImageNet than on CIFAR. The overfitting issue is not that severe on ImageNet. Hence, our binarization method could be more suitable for relatively data-deficient tasks.

### 3.4 ABLATION STUDY

We conduct a series of ablative studies of our proposed method in CIFAR-10/100 and ImageNet datasets with the ResNet18 architecture. By adjusting the coefficient $\lambda$ in the loss function $\mathcal{L}_{CMIM}$ (Eq. 11), where $\lambda = 0$ equals to no CMIM loss are added as our baseline. In the ablative studies, we introduce IR-Net [8] as our baseline method on all the datasets. The results are shown in Table 3. With $\lambda$ increasing, the performance improvements show the effectiveness of CMIM loss.

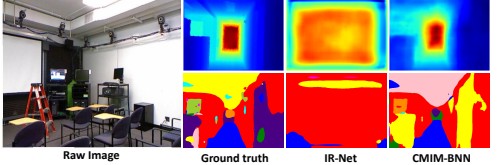

**Raw Image**    **Ground truth**    **IR-Net**    **CMIM-BNN**

Figure 5: Results of depth estimation and segmentation.

Table 3: Ablation study of CMIM. The results are presented in the form of accuracy rate (%). $\lambda = 0$ denotes no CMIM loss added, serving as our baseline.

| Dataset | $\lambda$ 0 (baseline) | 0.2 | 0.4 | 0.8 | 1.6 | 3.2 | 6.4 | 12.8 |
|---|---|---|---|---|---|---|---|---|
| CIFAR-10 | 87.59 | 90.92 | 91.63 | 92.06 | **92.18** | 91.89 | 91.32 | 91.01 |
| CIFAR-100 | 64.53 | 68.21 | 69.31 | 70.67 | 70.86 | 71.09 | **71.19** | 71.17 |
| ImageNet-1K | 58.03 | 59.29 | 59.99 | **61.22** | 61.17 | 61.02 | 60.64 | 59.7 |

### 3.5 GENERALIZATION ABILITY

To study the dependence of the binary activations from the same layer, we visualize the correlation matrix of those activations by using the shade of the color to represent the cosine similarity of two activations. Red stands for two activations are similar and blue *vice versa*. As shown in figures 4a-4d, CMIM captures more intra-class correlations (diagonal boxes are redder) and alleviates more inter-class correlations (non-diagonal boxes are bluer). Those intensified representative activations are constructive for fine-tuning down-stream tasks. To further assess the generalization capacity of the learned binary features, we transfer the learned binary backbone to the image segmentation and depth estimation on NYUD-v2 dataset. We follow the standard pipeline for fine-tuning. A prevalent practice is to pre-train the backbone network on ImageNet and fine-tune it for the downstream tasks. Thus, we conduct experiments with DeepLab heads with binary ResNet18 backbone. While fine-tuning, the learning rate is initialized to 0.001 and scaled down by 10 times after every 50K iterations and we fix the binary backbone weights, only updating the task-specific heads layers. The results are presented in Figure 4h, X-axis is the depth estimation accuracy (-logrmse, higher is better), Y-axis is segmentation performance (mIoU, higher is better) and the size of dot denotes the performance of classification (bigger is better). And the visualization samples are presented in Figure 5. We can witness that the models with backbone pre-trained by CMIM outperform other methods on both segmentation and depth estimation tasks.

## 4 RELATED WORK

In [7], the researchers initiate the studies of BNNs by using the sign function to binarize weights and activations to $\pm 1$. To eliminate the vanishing gradient issue caused by the sign function in the binarization, the straight-through estimator (STE) [44] is utilized for the network backpropagation. Based on this archetype, copious studies contribute to improving the performance of BNNs. For example, researchers in [9] disclose that the quantization error between the full-precision weights and the corresponding binarized weights is one of the major obstacles degrading the representation capabilities of BNNs. Reducing the quantization error thus becomes a fundamental research direction to improve the performance of BNNs. Researchers propose XNOR-Net [9] to introduce a scaling factor calculated by L1 norm for both weights and activation functions to minimize the quantization error. Inspired by XNOR-Net, XNOR++ [38] further learns both spatial and channel-wise scaling factors to improves the performances. Bi-Real [11] proposes double residual connections with full-precision downsampling layers to mitigate the excessive gradient vanishing issue caused by binarization. ProxyBNN [36] designs a proxy matrix as a basis of the latent parameter space to guide the alignment of the weights with different bits by recovering the smoothness of BNNs. Those methods have advanced the network binarization techniques.

Nevertheless, we argue that those methods focusing on narrowing down the quantization error and enhancing the gradient transmission reach their bottleneck (*e.g.* 1W32A ResNet-18 trained by ProxyBNN achieves 67.7% Top-1 accuracy on ImageNet, while full-precision version is only 68.5%). Because they neglect the activations in BNNs, especially the relationship between the binary and latent full-precision activations. We treat them as discrete variables and investigate them under the metric of mutual information. By maximizing the mutual information, the performance of BNNs is further improved.

## 5 CONCLUSION

In this paper, we investigate the activations of BNNs by introducing mutual information to measure the distributional similarity between the binary and latent full-precision activations. We take advantage of the exclusive structure of the BNN, where the binary and real-valued networks exist on the same forward pass, and establish a proxy contrastive learning task to maximize the targeted mutual information. We name this method **CMIM**-BNN. Because of the push-and-pull scheme in the contrastive learning, the BNNs derived by our method have better representation ability, benefiting downstream tasks, such as classification and segmentation, *etc.*. We conduct experiments on CIFAR-10/100, ImageNet (for classification) and NYUD-v2 (fine-tuning for depth estimation and segmentation). The results show that CMIM outperforms several state-of-the-art binarization methods on those tasks.

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
