# OpenReview forum: "Contrastive Mutual Information Maximization for Binary Neural Networks"
_ICLR.cc/2022/Conference — ICLR 2022 Submitted_

### Official Review · Reviewer_vKaa · 2021-10-19

**Correctness:** 3
**Technical Novelty And Significance:** 3
**Empirical Novelty And Significance:** 2
**Recommendation:** 3
**Confidence:** 5

**Details Of Ethics Concerns:**

I have carefully read the current manuscript and confirm the direct similarity between the submission and "Contrastive Representation Distillation" (ICLR2020) paper. Though the authors claim that they apply the contrastive loss to BNN (binary neural networks) training, the core part of the derivation (e.g., Equation(4-10)) remains the same as "Contrastive Representation Distillation" (e.g., Equation(4-19)).
Besides, I only found a reference to the ICLR20 paper in the middle of page 4 without mentioning the direct similarity.

Equations in Paper385 | Equations in ICLR20 paper
------------ | -------------
 Eq.(4) | Eq.(6)
Eq.(5) | Eq.(7)
Eq.(6) | Eq.(8)
Eq.(7) | Eq.(9)
Eq.(8) | Eq.(19)
Eq.(9) | Eq.(10)
Eq.(10) | $h^*(T,S)$ in Eq.(12)


**Main Review:**

pros.
1. The experiment results reported in Table 1 and Table 2 seem quite strong.
2. The idea of applying contrastive loss to BNN training seems new. It is interesting to see that full-precision latent weights can still be used in a new manner.

cons.
1. The main concern is the direct similarity between CMIM and CRD [1]. Please refer to the "Details Of Ethics Concerns".
2. Since the core part of this manuscript shares the same idea with CRD, the authors may overclaim the contribution of "a novel CMIM framework". In light of this, I think the novelty of this paper can be limited.
3. Though introducing latent weights into BNN training seems new, the extra training cost remains the same as the standard Knowledge Distillation (KD) framework based on FP32 teacher networks. However, I found no ablation study on it. Why CMIM should outperform CRD (if the authors argue that CMIM is indeed different from CRD)?
4. It is well known that KD should further improve the performance of student networks. Besides, Label Refinery [2] and Real-to-Binary [3] have shown that KD+BinConv leads to $+7$% Top-1 accuracy on ImageNet with XNOR-Res50 and $+4.3$% Top-1 accuracy on CIFAR-100 with Res18.
5. Since CMIM achieves much higher accuracy than RBNN, it is no doubt that t-SNE results will be improved. Note that the authors tend to maximize the lower bound of the mutual information. I expect some in-depth analysis on the change of mutual information during training.
6. How to prove that $h^*(\mathbf{a}^{k,i}_B,\mathbf{a}^{k,j}_F)=P(i=j|\mathbf{a}^{k,i}_B,\mathbf{a}^{k,j}_F)$?
7. Is there any constraint on the form of $h^*(\mathbf{a}^{k,i}_B,\mathbf{a}^{k,j}_F)$? Why use Eq.(8) as the critic function to approximate the target distribution? How to determine whether $h^*(\mathbf{a}^{k,i}_B,\mathbf{a}^{k,j}_F)$ converges to the target distribution?
8. Strangely, the most related work CRD is not included in section "Difference with other contrastive learning methods."

ref:
* [1] Contrastive Representation Distillation. ICLR2020
* [2] Label Refinery: Improving ImageNet Classification through Label Progression. arXiv2018
* [3] Training Binary Neural Networks with Real-to-Binary Convolutions. ICLR2020

**Summary Of The Paper:**

The authors propose to make full use of the full-precision latent weights in BNN training by utilizing the popular contrastive loss between samples generated by full-precision activations and binary counterparts. They follow the derivations of "Contrastive Representation Distillation" to bridge the gap between mutual information maximization and the proposed loss function. The experiment results show consistent improvements over strong baseline methods on image recognition tasks.

**Summary Of The Review:**

The authors may clearly discuss the differences between CMIM and existing works. The current draft makes it hard to fully evaluate the contributions of this paper.

---

> ### Author Response · Authors · 2021-11-22
> **Respond to Reviewer vKaa**
>
> Thank you for your thoughtful comments.
>
> The novelty of our method, we elaborate on the differences between our work and [CONTRASTIVE REPRESENTATION DISTILLATION, ICLR 2020] in the to-all-reviewers response. Most of your concerns are covered in this response.
>
> As for the experiments, the two works mentioned by you do not release their codes. In addition, we have compared ours with several code-released SoTA methods and performed ablative studies via adding our designed module on those methods to comprehensively investigate the effectiveness of CMIM.

---

### Official Review · Reviewer_LrbF · 2021-10-26

**Correctness:** 3
**Technical Novelty And Significance:** 2
**Empirical Novelty And Significance:** 2
**Recommendation:** 3
**Confidence:** 5

**Main Review:**

**Strengths**

* The paper has good writing and structure, thus easy to follow.
* The paper demonstrates promising experimental results. The results show that the proposed method improves performance on different data sets, multiple models, and multiple tasks. It shows outstanding scalability and generality.

**Weaknesses**

* **Limited Novelty.** However, the contribution and novelty of the proposed approach are limited. The approach looks like applying the method proposed in [1] to the BNN optimization problem. The combination is very straightforward and lacks real innovation.
* **A convincing explanation is required.** The effect of data augmentation by utilizing negative pairs seems to be more effective, but the mutual information maximization between positive pairs lacks a convincing explanation. How do we understand the essential meaning of maximizing mutual information between fp activation and their signs? So let P denote the distribution of fp activation and Q denote the distribution of their signs (so-called binary activation in the paper). The mutual information of P and Q is the entropy of Q, since Q is always part of P. The increase of the entropy of P cannot explain the increase of the mutual information. Only when the distribution [-1, 1] of Q itself is closer to 50%:50%, the entropy is the largest. In this case, the binarization is more uniform, and it can encode more information. But in this case, mutual information maximization has the same effect as maximizing entropy(Q).
I hope the author can provide reasonable proof or at least give a more convincing explanation of why maximizing mutual information of P and Q makes sense? At present, it feels a bit rough to apply the existing contrastive learning framework to the BNN optimization problem.
* **AC please note that** page 5 after equation 10: “where the detailed proof is shown in the supplementary material” I am very confused, where is the author’s proof? I did not find any supplementary materials. In addition, the formula derivation in this paper has many similarities with [1]. The supplementary materials in [1] provide specific derivations of the similar formula . . .
* **Real difference with other contrastive learning methods not found.** I can't see the obvious difference from other contrastive learning methods. It seems to be the application of existing methods in the BNN training scene.
* **Unconvincing experimental results**: Experimental Setup on ImageNet is quite confusing. Hyperparameters are pretty different from the known open source BNN methods, including the use of weight decay (the essence of BNN training is to change the sign of weight, weight decay is almost useless, so usually wd is not used), too large initial learning rate (usually 10-100 times smaller), SGD optimizer (usually adam, see AdamBNN paper for explanation), I am curious how the author can exceed the results of ReActNet by training only 100 epochs because of the latter trains at least 256 epochs. In summary, according to the above questions, I am curious whether the author considers providing code and more experimental details such as logs and models? This will be a great help.

**Minor issue**

* How do you implement BATS? Since the author still didn’t share the codes as they claimed. Would you like to share your implementation? It will be a great effort.
* The ImageNet results using ResNet18 (also ResNet34) seem to be relatively weak. Strong baselines such as ReActNet with BiReal backbone are not compared (65.9% top1 accuracy, https://github.com/liuzechun/ReActNet/). Other stronger results from RealToBinary and MeliusNet are also ignored (All of them achieved a higher accuracy with aligned computation complexity and model size).
* Related work overlooked a lot of recent efforts on binary neural network research, e.g., ReActNet, MeliusNet, AdamBNN, etc. However, some of them are compared in the experiment section.

[1] Yonglong Tian, Dilip Krishnan, Phillip Isola, Contrastive Representation Distillation,  ICLR2020


**Summary Of The Paper:**

This paper proposes an auxiliary method for training BNN models. It follows the idea of contrastive-mutual-information-maximization, which utilizes the full-precision and the binary activation of BNN to form positive (binary and fp activation of the same sample) and negative (binary activation of different samples) pairs for contrastive training. The auxiliary contrastive loss can provide data augmentation and effectively enhance the model's generalization ability.

**Summary Of The Review:**

Regarding the existing problems in the paper, I recommend rejection for now, but it may be adjusted according to the rebuttal.

---

> ### Comment · Area_Chair_GLir · 2021-11-05
> **comment**
>
> Thanks for looking at the technical details. Indeed there is no supplementary.
> In the first quick reading, I got stuck already at (4): I could not interpret $i=j$ as a random variable because $i$ and $j$ are enumerated exhaustively to produce $KN^2$ pairs.
>
> Could the authors clarify?

---

> > ### Author Response · Authors · 2021-11-23
> > **Respond to AC GLir**
> >
> > In our content, $i == j$ can be treated as a variable. If $i = j$, return 1; else, return 0. Intuitively, for the activations of each layer, their $1*N$ positive pair and $N*(N-1)$ negative pairs. Note that the activations are layerwise and not from a teacher-and-student pipeline as we discussed in the to-all-reviewers response.

---

> ### Author Response · Authors · 2021-11-23
> **Respond to Reviewer LrbF**
>
> Thank you for your thoughtful comments.
>
> The novelty of our method, we elaborate on the differences between our work and [CONTRASTIVE REPRESENTATION DISTILLATION, ICLR 2020] in the to-all-reviewers response. Most of your concerns are covered in this response.
>
> As for the experiments, we basically utilize the code base of RBNN, which is the code-available SoTA within ResNet-18 architecture. In the literature of network binarization, most works implement their proposed methods with ResNet-18 and ResNet-34. Admittedly, we forget to mention that when we implement our method with the setting ReActNet+CMIM, we use the codebase of ReActNet. However, we still think that our experiments which are mainly based on ResNet and complementarily on invariant ResNet (which is rarely studied in the content of binarization) such as ReActNet and BATS, are comprehensive enough to demonstrate the effectiveness of our method. Since it is common in most binarization SoTA work
>
> Moreover, the code for BATS can be found in https://github.com/1adrianb/binary-networks-pytorch, which is a general codebase for binarization algorithms (not released by the authors, at least useable).

---

### Official Review · Reviewer_iEui · 2021-10-30

**Correctness:** 3
**Technical Novelty And Significance:** 3
**Empirical Novelty And Significance:** 3
**Recommendation:** 6
**Confidence:** 4

**Main Review:**

The idea of treating binary activations and real-valued activations as two views of the same image in self-supervised learning is interesting and makes sense. The results also well support the effectiveness of the proposed method.

Overall, the submission is technically sound. I have several questions as follows:

1. Does inserting the MLP layers and using the proposed CMIM module slow down the training greatly? Can the authors provide the training time comparison with the baseline method?

2. Have the authors visualized the binary and real-valued activation distributions inside the BNNs trained with CMIM? Does it look significantly different from other binary neural networks?

Minor issues:

1. The authors mentioned researchers in [13](CVPR 2021) propose to shift the thresholds of binary activation functions. However, learning to shift the thresholds in binary neural networks is first proposed in ReActNet (ECCV 2020).

2. The caption in Fig. 4: “the effect of number of negative samples in contrastive mutual information maximization (c-f)” should be “(e-f)”

**Summary Of The Paper:**

This paper proposed an approach to reduce the performance gap between the binary neural networks and their real-valued counterparts via maximizing the mutual information between the binary activations and the real-valued activations that exist in the binary neural network. Specifically, the authors propose to treat binary activations and real-valued activations as two views of the same image and use contrastive learning to pull these positive pairs while pushing other negative pairs generated from different images.

**Summary Of The Review:**

In summary, this is an interesting paper with clear motivation and novel techniques. I will raise my score if my concern about the training time is well addressed.

---

> ### Author Response · Authors · 2021-11-22
> **Respond to Reviewer iEui**
>
> Thank you for your thoughtful comments.
>
> As we talked about in the method part and to-all-review parts, the contrastive objective is a sophisticated tool for realizing mutual information.  Thus, in practice, the large number of negative samples hinder the implementation of our designed algorithm, as the cost of memory would be expensive if we operate all the data augmentations (e.g., 16,384 on ImageNet) on the samples simultaneously. To overcome this problem, we leverage the Memory Bank technique [1] to iteratively update and acquire the representations of samples with augmentations from the Memory Bank, as works [1,2,3] in contrastive learning applied. Due to this technique, the required memory only increases 10% and training time 20% compared with the baseline method.
>
> We do visualize the binary and real-valued activation distributions with bar charts inside the BNNs trained with our method, but we do not observe any obvious difference. Our speculation is that mutual information is a high-level property to describe distributions. The bar chart cannot reflect the difference. However, the s-SNE in Fig.3 can partially reflect the distributions of activations, where ours is better than others.
>
> [1] Representation learning with contrastive predictive coding, Oord et. al., Arxiv, 2018
>
> [2] Learning deep representations by mutual information estimation and maximization, Hjelm et. al., ICLR, 2019
>
> [3] Learning representations by maximizing mutual information across views, Bachman et. al., NeurIPS, 2019

---

### Official Review · Reviewer_9r7Z · 2021-11-02

**Correctness:** 3
**Technical Novelty And Significance:** 2
**Empirical Novelty And Significance:** 2
**Recommendation:** 5
**Confidence:** 4

**Main Review:**

My main concern is about the novelty. A similar contrastive distillation method has already been proposed in [1]. This submission falls into the first setting shown in Figure 1(a) of [1], which compresses the model by maximizing the mutual information between the binary student network and the full-precision teacher network. The derivations of contrastive mutual information maximization in equations (4)-(7) are almost the same as (4)-(11) in [1]. The loss function (9) in this paper also resembles that (equation (18)) of [1]. The authors should clarify the connections/differences of this submission with [1].

Minors:
- What does \dot mean in equation (1)?
- Figure (4) legend should be CMIM instead of MIM.

[1] CONTRASTIVE REPRESENTATION DISTILLATION, ICLR 2020.

**Summary Of The Paper:**

This paper proposes to use contrastive distillation to train a binary network by maximizing the mutual information between itself (student network) and the full-precision network (teacher network). Empirical results show that this new objective further improves the binarization performance on top of several recent binary networks on image classification tasks. The authors also empirically show that models trained with the proposed contrastive objective have good transfer performance.

**Summary Of The Review:**

This paper is overall structured clearly and well written. However, the novelty may be limited as it can be viewed as an application of the previous contrastive distillation method to the quantization task.

---

> ### Author Response · Authors · 2021-11-22
> **Respond to Reviewer 9r7Z**
>
> Thank you for your thoughtful comments.
>
> We elaborate on the differences between our work and [CONTRASTIVE REPRESENTATION DISTILLATION, ICLR 2020] in the To-all-reviewers part.
>
> The \dot in Eq.1 means the linear multiplication in a neural network.
>
> We will modify the typo.

---

> > ### Comment · Reviewer_9r7Z · 2021-11-29
> > **Response to authors**
> >
> > I thank the reviewers for the detailed comment. However, I still think this paper is better viewed as an interesting application of the CRD to the area of network quantization, rather than as introducing a fresh new algorithm. Besides, Eq.(4-9) are not common to all contrastive learning methods, and should be credited to the original CRD paper which first introduces them.

---

### Author Response · Authors · 2021-11-22
**To all the reviewers: Comparisons to Contrastive Representation Distillation**

We notice that three reviewers (R1, R2, and R4) argue our work’s novelty, requiring specific comparisons to CRD [6]. We would like to discuss the differences between the two works.

Firstly, we would like to highlight that our work can not be seen as a teacher-and-student backbone. In KD, the teacher is basically fixed to offer additional supervision signals and is not optimizable. But in our formulation, we take advantage of the exclusive structure in BNN, where full-precision (FP) and binary (after binarization) activations exist in the same forward pass i.e, only one network (BNN) is involved, without using another network as a teacher. Therefore, the accuracy improvement of the BNN trained by our method is purely benefited from the activation alignment in a contrastive way, rather than a more accurate teacher network. Apart from this, instead of only using the activation of the final layer, we align the activations layer-by-layer with a hyperparameter to adjust the weight of each layer as shown in Eq.4-11 in our paper, which is a more suitable design for BNN.

Secondly, both works are motivated by mutual information. But both works never highlight mutual information with contrastive learning as their novelty. Using contrastive objective as a tool to realize mutual information maximization is not originally proposed by our work either CRD. In contrast, the selling points of ours and theirs are to introduce contrastive learning to solve the targeted problems in task-specific ways. Actually, many recent self-supervised learning methods are based on the idea of maximizing mutual information. For instance, Contrastive Predictive Coding (CPC) [2], Deep InfoMax (DIM) [3], and Augmented Multiscale DIM (AMDIM) [4]. These methods are based on noise contrastive estimation (NCE) back to 2010[1] which, under specific conditions, can be viewed as a bound of MI. In these paper, researchers introduce flexible parametric distributions or critics parameterized by neural networks that are used to approximate the densities $(p(y), p(y\mid x))$ or density ratios $(p(x\mid y)/ p(x) =p(y\mid x)/ p(y))$ [5]. In deep learning background, the resulting objective functions are commonly named InfoNCE.

Thirdly, as for the similar equations, they are also shown in those fundamental works. Again, we are the first on introducing contrastive learning in the BNN training process from the perspective of mutual information maximization and achieve promising accuracy gain.

Ours | NCE[1] |CPC[2]| description
------ | ------ | ------ | ------
Eq.4 |  Eq.7  | |  Bayes' law
Eq.5 |  Eq.6  | Eq.5 | assumption of independency
Eq.6 |           | |  definition of logarithm
Eq.7 |            |    Equation below Eq.5.   |    definition of expectation
Eq.8 |  Eq.4,5 |  Eq. 2,3 | standard critics in contrastive learning
Eq.9 | Eq.10,11  |    |

Note that we refer to all those aforementioned studies in our manuscript.

[1] Noise-contrastive estimation: A new estimation principle for unnormalized statistical models, Gutmannet et. al., AISTAT, 2010
[2] Representation learning with contrastive predictive coding, Oord et. al., Arxiv, 2018
[3] Learning deep representations by mutual information estimation and maximization, Hjelm et. al., ICLR, 2019
[4] Learning representations by maximizing mutual information across views, Bachman et. al., NeurIPS, 2019
[5] On Variational Bounds of Mutual Information, Ben Poole et, al., ICML 2019
[6] Contrastive representation distillation, Tian et. al., ICLR 2020

---

> ### Comment · Reviewer_vKaa · 2021-11-23
> **Ethics Concerns still exist**
>
> - Though the authors claim that "we refer to all those aforementioned studies in our manuscript", I wonder why the MOST related work CRD was mentioned only once? The authors clearly show the similarity between the proposed method and NCE/CPC, then why exclude CRD from the comparison?
> - The authors seem to indicate that Eq.(4-9) are ALL common sense in contrastive learning, which I can not agree with. The derivation of Eq.(4-10) as a whole was first proposed in CRD, not one of the equations.
> - Besides, the authors claim that they have included a proof on $h^*(\mathbf{a}_B,\mathbf{a}_F)=P(i=j|\mathbf{a}_B,\mathbf{a}_F)$ in Appendix (unfortunately it was not provided, as pointed out by Reviewer LrbF), coincidentally, which has been included in CRD's Appendix.

---

> > ### Author Response · Authors · 2021-11-23
> > **The paper drafting criterion.**
> >
> > Because of the space limitation, we should focus on the most important works related to our work. In addition, as we stated, our work is not a teacher-and-student pipeline, there are several differences between our setting and knowledge distillation. Thus, from the perspective of mutual information maximization with contrastive objectives, of course, we need to compare with the most original works, which are the works we mentioned in our manuscript.

---

### Decision · Program_Chairs · 2022-01-20

**Decision:**

Reject

**Comment:**

## Description

The paper applies ideas from contrastive representation learning to train binary neural networks. Namely, the algorithm promotes binary representations to be similar to the full-precision representations while at the same time it promotes binary representations to be dissimilar from full-precision representations corresponding to other input images. This is enforced for activations in all layers by the added contrastive loss (9).

## Decision

The main weakness of the paper pointed by reviewers were 1) overlap of the large part of derivation with the prior work [25] Tian et al. "Contrastive representation distillation", ICLR 2020; and 2) the meaning of the derivation when applied in the setting of the paper to binary and full precision weights and its soundedness. The authors proposed their arguments for 1). The reviewers board considered these arguments and did not agree (see below). Point 2) was not addressed by authors (no paper revision, justifications, proofs corresponding to the missing supplementary). It was discussed further and was found critical (see below), such that it is a clear reason for rejection regardless of 1). Overall, the idea is interesting and the method appears to be helpful experimentally, however the paper needs a major revision that would address the two points.

## Details

### Overlap with CRD

Reviewers were in a consensus on this issue, disagreeing with authors. Since the whole derivation chain of the contrastive loss already exists in the CRD work [25], it is redundant to repeat this derivation if not raising ethical concerns. Instead an original work should review or just refer to the existing derivation and only discuss the new context and e.g. change the critic function $\hat h$.

### Meaning of the derivation

The reviewers have questioned the soundness of the initial criterion of MI between binary and full precision activations, as it reduces to just the entropy of binary activations. In particular, it seems very different in meaning to the contrastive loss the paper optimizes in the end. Here is additional feedback from the discussion.

1. Maximizing the entropy of binary activations with respect to the data distribution makes some sense. If a single binary activation was considered, its entropy is maximized when it is in the state 1 exactly for 50% of the data. Which makes it discriminative of the input. A similar centering can be achieved by Batch Normalization put in front of the activation -- if the preactivation distribution was symmetric, then BN would achieve the max entropy for the sign of preactivation. Such network design is not uncommon.
Maximizing the entropy of the full vector of binary activations appears more difficult. However we can also understand it as the mutual information between the input image and the layer of binary activations. Thus the criterion is to retain as much information about the input as possible. This makes sense as a regularization (often neural networks are regularized by adding data reconstruction capabilities / loss), and is aligned well with goals such as re-using the features for other tasks (as in Sec .3.5) but contradicts to some other principles proposed in the literature, e.g. the information bottleneck (that the maximum information about the target rather than the input should be preserved).
Amongst methods that study the direction of maximizing the entropy in binary networks, reviewers mention IR-Net and Regularizing Activation Distribution for Training Binarized Deep Networks. The architecture with BN before activation is used in the latter work and some more recent works, e.g. BoolNet.

2. It is not clear whether optimizing the contrastive loss retains the same meaning as maximizing MI. The derivation from CRD paper used here applies several lower bounding steps. Maybe the strongest one is that the critic is chosen to be of a specific function rather than a universal approximator. However there is no obvious gap. In fact knowing that binary activations are just a sign mapping of full precision ones, should allow one to estimate $p(i=j| a^i_B, a^j_F)$ in a simple way.

3. In the estimator $h$ in (8) the authors make a mistake (applying their and CRD theory incorrectly):
$h$ should be the probability of a conditional Bernoulli variable estimating $p(i=j| a^i_B, a^j_F)$. It should not depend on $a^j_F$ for other values of $j$ than the given one. However in the denominator in (8) it does. Therefore this estimator, and as a result the specific NCE loss proposed, appear unjustified. If the critic from CRD eq. (19) is adopted, it is not clear whether it makes sense for a pair of binary and full precision descriptors (note that for $i=j$ the scalar product between the two is just $\|a_F\|_1$).  It seems that the design of a meaningful critic is a serious gap the authors should address. Observing that the initial objective, the MI criterion, was in fact independent of full-precision states (as it is the entropy of binary states), one can propose that an appropriate critique should use binary states only, such as
$$
h(a_B^i,a_B^j) = \sigma(\left<a_B^i, a_B^j\right>  + c ).
$$
When fixing $\hat h$ the result in (10) that the maximum likelihood estimator for $p(i=j | a_B^i, a_F^j)$ with a generic neural network can approximate this distribution arbitrary well becomes irrelevant.

When the paper speaks of randomness, e.g. "binary and full precision activations as random variables, considering "i=j" as a random variable, it is needed to specify the source of randomness or the distribution, i.e. to add "for a network input drawn from the data distribution" in the first case and "under i and j picked at random uniformly in the batch" in the second.

Theoretically, the paper would become more convincing, if the the entropy of binary activations was measured by independent tools from the literature after training with and without NCE loss and it was shown that indeed the method achieves an improvement in this objective, reconfirming that the principle and the derivation were sound. An ablation study on other modifications such as weight decay may be helpful to convince researchers that the main source of improvements in experiments is the new contrastive loss. Note that not all reviewers were convinced by current experimental results due to lack of descriptions / code to fully reproduce and or lack of such ablation studies.